# *Staphylococcus aureus*—An Additional Parameter of Bathing Water Quality for Crowded Urban Beaches

**DOI:** 10.3390/ijerph18105234

**Published:** 2021-05-14

**Authors:** Nancy Topić, Arijana Cenov, Slaven Jozić, Marin Glad, Diana Mance, Dražen Lušić, Damir Kapetanović, Davor Mance, Darija Vukić Lušić

**Affiliations:** 1Department of Environmental Health, Teaching Institute of Public Health of Primorje-Gorski Kotar County, Krešimirova 52a, 51000 Rijeka, Croatia; nancy.topic@gmail.com (N.T.); arijana.cenov@zzjzpgz.hr (A.C.); marin.glad@zzjzpgz.hr (M.G.); darija.vukic-lusic@zzjzpgz.hr (D.V.L.); 2Laboratory of Microbiology, Institute of Oceanography and Fisheries, P.O. Box 500, 21000 Split, Croatia; 3Department of Physics, University of Rijeka, Radmile Matejčić 2, 51000 Rijeka, Croatia; diana.mance@uniri.hr; 4Department of Environmental Health, Faculty of Medicine, University of Rijeka, Braće Branchetta 20, 51000 Rijeka, Croatia; drazen.lusic@medri.uniri.hr; 5Faculty of Health Studies, University of Rijeka, Viktora Cara Emina 5, 51000 Rijeka, Croatia; 6Ruđer Bošković Institute, Division for Marine and Environmental Research, Bijenička Cesta 54, 10000 Zagreb, Croatia; kada@irb.hr; 7Faculty of Economics, University of Rijeka, Ivana Filipovića 4, 51000 Rijeka, Croatia; davor.mance@efri.hr

**Keywords:** bathing water quality, crowded beaches, fecal indicator bacteria, *Staphylococcus aureus*

## Abstract

During the last years, the report of the occurrence of waterborne disease symptoms related to non-enteric pathogens has increased, without any record of higher levels of indicator bacteria (*Escherichia coli* and intestinal enterococci). Therefore, the use of current indicators is not always adequate when assessing the overall potential health risk and the inclusion of additional parameters needs to be examined. This paper reports on the incidence and levels of *Staphylococcus aureus* at 258 locations in Primorje-Gorski Kotar County (Croatia) recorded by official bathing water quality monitoring, as well as supplemental monitoring carried out at the two most frequented beaches in the City of Rijeka. The number of bathers was found to be the main factor affecting *S. aureus* levels (r = 0.321, *p* < 0.05). The share of *S. aureus* positive samples from the official monitoring was significantly lower, when compared to the share of samples from supplemental monitoring (2.2% and 36.3%, respectively; *p* < 0.01). Besides the number of bathers, one of the main reasons is likely the higher sampling frequency. No correlation was found between *S. aureus* levels and the indicator bacteria. The results indicate that the determination of *S. aureus* and increased sampling frequency is recommended for overcrowded beaches.

## 1. Introduction

The management of bathing waters in the European Union is regulated by Bathing Water Directive 2006/7/EC (BWD) [1]. Unlike the first edition of the Directive, 76/160/EEZ, which had defined a large set of microbiological, chemical, and physical parameters that bathing water had to meet [2], a new Directive defines only two fecal indicator bacteria (FIB), *Escherichia coli* and intestinal enterococci, as the parameters in routine monitoring of coastal bathing water quality. According to BWD, the European Commission is required to review the current Directive, no later than 2020, with particular reference to World Health Organization (WHO) recommendations [1]. After a thorough analysis of available scientific research, WHO concluded that there is currently no scientifically substantiated basis for the introduction of new parameters for microbiological monitoring of bathing water quality. Therefore, the WHO recommended that the use of current indicators should be continued [3]. This recommendation is based on the results of the most relevant studies discussed in the report, where the relationship between FIB levels and incidence of the most commonly reported illnesses, such as gastrointestinal illness and skin symptoms, was studied and recorded.

Nonetheless, there is a significant number of studies reporting on the incidence of waterborne diseases associated to bathing, whereby FIB levels did not indicate a risk of non-enteric bacterial infections [4,5]. In a review by Korajkic et al. [6], the absence of a correlation between observed illness and FIB levels was reported in seven studies out of 17 for enterococci, and four out of six studies for *E. coli*.

Since it is obvious that the correlation of indicator bacteria with particular pathogens and/or illness occurrence is still being questioned, and the enumeration of current indicators may be insufficient to assess the risk of non-enteric pathogens [7,8], some bacteria such as *Staphylococcus aureus* are suggested as potential additional parameters for monitoring of coastal bathing water quality [7,9,10].

*S. aureus* is considered one of the most resistant non-sporogenic bacteria, with the ability to survive high temperatures, drying, extreme pH, high salinity, antibiotics, and disinfection treatments [11,12,13,14]. These bacteria are normal commensals of the human nasopharynx, anterior nares, perineum, and skin [15]. Most of the time, these bacteria cause no problems or result in relatively minor skin infections, but occasionally the infections can turn deadly and cause serious health issues. It is estimated that 20–40% of the human population transmits this opportunistic pathogen [16,17,18]. Besides humans, domestic animals [19,20] and birds [21] are considered as an important reservoir of this bacteria. High levels of *S. aureus* shed by bathers (adults and children) of about 10^5^–10^6^ CFU/person during 15 min [22,23] and a positive correlation with skin, ear, and respiratory infections in seawater [24], suggest that this opportunistic pathogen is a possible additional parameter to be considered in monitoring the seawater quality of crowded beaches. This is supported by higher adaptability of *S. aureus* to seawater conditions because of its high survival capacity at higher salinities, compared to FIB [25].

The main goal of this paper is to assess whether the inclusion of *S. aureus* in official (routine) monitoring of bathing water quality should be considered on the basis of the results of *S. aureus* incidence and levels in seawater under different environmental conditions, such as abiotic factors, beach load/number of bathers, and also FIB levels. 

## 2. Materials and Methods

### 2.1. Study Area and Sampling

During the 2017 bathing season, national official (routine) monitoring of bathing water quality in Primorje-Gorski Kotar County (Croatia) was carried out fortnightly, from mid-May to the end of September (10 samples per site per season). Samples were taken at 258 official bathing sites. Besides the mandatory indicator bacteria (*E. coli* and intestinal enterococci), an additional parameter, *S. aureus*, was included.

Additional, supplemental monitoring was carried out at four official bathing sites located along the two most frequented urban beaches of the county, in the west part of the city of Rijeka, namely, Ploče beach (east-PE and west-PW bathing sites) and Kantrida beach (east-KE and west-KW bathing sites). Unlike official monitoring, sampling was performed every day, including the weekends, with 248 samples taken during the period 1 July to 31 August 2017.

Kantrida beach is a 270-m long urban pebble beach with many beach facilities (coffee bars, showers, slides, benches, parking areas, and ancillary facilities) (Figure 1). Near the beach, there is an area characterized by numerous fresh water springs. The most important one is Cerovica spring, which significantly affects the quality of the seawater. Abundant rainfall causes short-term contamination of the sea, which increases the health risk for bathers [26].

Ploče Beach is a 330-m long urban pebble beach. It is located in front of the Kantrida Swimming Pool Complex, a sport, recreation, and entertainment complex (Figure 1). Ploče beach has been awarded Blue Flag status, an international recognition of exceptional cleanliness, quality, and tidiness of a beach.

Sampling for both the official and supplemental monitoring program was performed during the morning hours. The samples were taken by applying the aseptic technique, 30 cm below the surface, in water with a minimum depth of 1 m. All samples were processed as soon as possible, on the same day. In total, 2867 seawater samples were obtained during regular and supplemental monitoring, and tested for *S. aureus*. Sampling reports were produced throughout the official and supplemental monitoring program, including the sample identification number of each location, the sampling date and time, and air and sea water temperature. Beach load data (number of bathers/children in diapers/dogs/seagulls) for each sampling event at each sampling site were determined by visual observation at the time of sampling. This was done by the students, by direct counting, or subsequent processing of the photographed situations. For ease of interpretation, the beach load data were subsequently classified into categories: *N* (0) = category “0”, *N* (1–50) = “1”, *N* (50–100) = “2”, *N* (100–150) = “3”, *N* (150–200) = “4”, *N* (>200) = “5”.

### 2.2. Sample Analysis

#### 2.2.1. Microbiological Analysis

Laboratory analysis was based on the detection and quantification of *E. coli* and intestinal enterococci, along with *S. aureus* as the additional parameter. All microbiological parameters were determined using a membrane filtration technique, the temperature modified HRN EN ISO 9308-1:2014 method [27,28] for *E. coli*, HRN EN ISO 7899-2:2000 for intestinal enterococci and the method described by Standard methods 23rd. Ed 2017. 9213 B for *S. aureus* [29]. In brief, for each parameter, the samples (10 mL and 100 mL) were filtered through 47 mm in diameter cellulose nitrate membranes, 0.45 μm pore size for *E. coli* and intestinal enterococci, and 0.22 μm for *S. aureus*. After filtration, the funnels were rinsed twice with sterile deionized water, and the membranes were then transferred to Chromogenic Coliform Agar (CCA) for *E. coli*, Slanetz & Bartley agar for intestinal enterococci and Baird-Parker agar for *S. aureus*.

CCA was incubated for 4 h at 36 ± 2 °C, followed by 20 h at 44 ± 0.5 °C. All dark-blue to violet colonies were counted as confirmed *E. coli*.

After incubation on Slanetz & Bartley agar at 36 °C ± 2 °C for 44 ± 4 h, the membranes used for the enumeration of intestinal enterococci were transferred to prewarmed (44 °C) Bile Aesculin Azide Agar and incubated at 44 ± 0.5 °C for 2 h. All pink, red, or brown colonies that developed a brown or black halo on Bile Aesculin Azide Agar were counted as confirmed enterococci.

Baird-Parker agar was incubated at 35 ± 0.5 °C for 48 ± 4 h. Staphylococci typically form slate-grey to jet-black smooth, entire colonies. The presence of presumptive *S. aureus* colonies was observed by a zone of egg yolk clearing when the membranes were raised from the medium. Depending on the total number of colonies on the membrane, all colonies (if there were less than 10 colonies) or at least 10 differentiated presumptive colonies from each membrane were verified by coagulase production and catalase reaction. All coagulase and catalase positive colonies were counted as confirmed *S. aureus*.

#### 2.2.2. Physical/Chemical Analysis

Air and seawater temperature were measured in situ, using a centigrade mercury scale thermometer (scale 0.1). The pH values and salinity were determined at the laboratory, using a pH meter and a conductometer, respectively. Likewise, seawater turbidity was estimated at the laboratory using an optical instrument—turbidimeter (nephelometric turbidity examination).

#### 2.2.3. Data Analysis

Normality of data was tested using the Kolmogorov-Smirnov test. Since the data, for all measured parameters, failed the expectation of Gaussian distribution, the following non-parametric tests were performed: Spearman’s rank order correlation, Chi-square test, Wilcoxon matched pairs test, Mann-Whitney U test, and Kruskal-Wallis H test. Principal component analysis (PCA) was performed to identify components that explain the major variation within data. For presentation of the results, descriptive statistical methods were applied (relative frequency, median, interquartile range-IQR), as well as graphs and tables. Statistical analysis of the data was performed using the Microsoft Excel Statistic Package (Redmond, WA, USA), Statistica 13 (Stat. Soft. Inc., Tulsa, OK, USA, SAD) and Canoco version 5 (http://www.canoco5.com/ (accessed on 07 May 2021)). The results were interpreted at a statistical significance level of under 0.05.

## 3. Results

### 3.1. Official vs. Supplemental Monitoring

The results revealed a higher share of samples positive for *S. aureus* for the supplemental monitoring program compared to national official monitoring (Chi2 test, *p* < 0.01). During official monitoring, only 2.2% of samples were positive for *S. aureus* (57/2619), while as many as 36.3% (90/248) of the supplemental monitoring samples were positive. Considering the results obtained for four sampling locations where supplemental monitoring was carried out, only 5% (1/20) of the official monitoring samples were positive during the same sampling period (1 July to 31 August 2017). The results of the Mann-Whitney U test indicate a significant difference (Z = 2.98; *p* < 0.01) in the medians of the *S. aureus* values in positive samples during official (median 2 CFU/100 mL, IQR 1–4 CFU/100 mL) and supplemental (median 4 CFU/100 mL, IQR 2–10 CFU/100 mL) monitoring.

Several groups can be discerned from the PCA biplot. The first consists of the number of bathers and children, air and seawater temperature and *S. aureus*, while the second group consists of *E. coli* and intestinal enterococci (Figure 2). Spearman’s correlation analysis supports the results of PCA analysis, showing a weak but statistically significant positive correlation between *S. aureus* and sea temperature (rs = 0.243; *p* < 0.05), air temperature (rs = 0.147; *p* < 0.05), number of bathers (rs = 0.321; *p* < 0.05), and number of children in diapers (rs = 0.203; *p* < 0.05). No significant correlation between *S. aureus* and FIB levels was found (Table 1).

### 3.2. Kantrida vs. Ploče Beaches

Although located in the west part of the city, on a small stretch of about 900 m, Kantrida and Ploče beaches showed significant differences in the percentage of samples positive for *S. aureus* (Chi2 test, *p* < 0.01), and also between the concentrations of *S. aureus* in positive samples (The Mann-Whitney test, Z = −3.73; *p* < 0.01). Considering the results of supplemental monitoring carried out on Kantrida and Ploče beaches, the percentage of samples positive for *S. aureus* was 49.2% (61/124) and 22.4% (29/124), respectively. The median value of the positive results obtained during supplemental monitoring was higher on Kantrida beach than on Ploče beach (Table 2). No statistically significant difference between the concentrations of *S. aureus* in positive samples was found between two sampling locations of the same beach, i.e., Kantrida East and West (Z = 0.533, *p* = 0.593) and Ploče East and West (Z = 0.044; *p* = 0.965).

Overall, of the total number of supplemental monitoring samples positive for *S. aureus,* 76.6% had a concentration of ≤10 CFU/100 mL (Figure 3). Out of all sample positive for *S. aureus*, in 18 samples (20%), the count of at least one of FIB was lower than the count of *S. aureus*, and in four samples, no FIB was detected.

Apart from the differences in the concentration of *S. aureus*, a difference in the concentrations of FIB was observed between the two beaches. The results of the Mann-Whitney U test indicate a significant difference in the concentration of routine fecal indicators, *E. coli* (Z = 6.67; *p* < 0.01) and intestinal enterococci (Z = 4.78; *p* < 0.01), between the beaches. Both, *E. coli* and intestinal enterococci concentrations were higher on Kantrida beach than on Ploče beach (Table 2).

Bathing water quality based on FIB data obtained during supplemental monitoring, and assessed using 95th percentile of all data, as defined by BWD, also showed better water quality on Kantrida beach when compared to the water quality on Ploče beach. Both sites on Ploče beach (PE and PW) were of “excellent” water quality, while water quality at one site on Kantrida beach (KW) was of “good” quality (Figure 4).

The Wilcoxon matched pairs test showed a significant difference (*p* < 0.05) in salinity and turbidity between Kantrida and Ploče beach, while no significant difference in temperature was recorded between these locations (Table 2).

During the 14 days of supplemental monitoring in 2017, the number of bathers at Kantrida beach was immense (N_bathers_ > 200; category = ”5”), especially on the west part of the beach. At Ploče beach, the highest recorded category was “4” (N_bathers_ = 150–200). The seagulls at Kantrida beach were classified into category “0” on 63 events, and 61 into category “1”, while at Ploče beach, 112 events were classified as “0” and only 12 as category “1” (Chi2-test, *p* < 0.01). Considering the witnessed number of dogs, there was only one event of category “1” at Kantrida beach, while all others were classified into category “0”. Regarding children in diapers, 148 events (53 on Kantrida and 95 on Ploče) belonged to category “1”, and 16 (Kantrida) to category “2”. All other events belonged to category “0”.

## 4. Discussion

The share of *S. aureus* positive samples at Kantrida and Ploče beaches from the official monitoring program was significantly lower (5%) compared to the supplemental monitoring program (36.3%). One of the main reasons is likely sampling frequency. The samples obtained from official monitoring of these two beaches were collected fortnightly, while the supplemental monitoring program sampling was performed on a daily basis, including weekends. With such a high sampling frequency, there is a higher probability of recording seawater contamination events. This is supported by the difference between official and supplemental monitoring as regards the share of FIB positive samples at Kantrida and Ploče beaches. That is, the share of samples that were positive for *E. coli* and enterococci taken at these beaches for official monitoring (biweekly sampling) was 71.8 and 70.0% respectively, while for supplemental monitoring (daily sampling), the share was 96.0 and 90.0%, respectively. This indicates that the sampling frequency could affect the probability of recording bathing water contamination events. Moreover, low sampling frequency has been recognized as the main cause of misclassification of bathing water sites [30]. According to the WHO assessment, an increase in sampling frequency from 10 samples per site per season to 20 would reduce misclassification from about 22% to about 14% [30]. Furthermore, the results showed a higher share of *S. aureus* positive samples on Kantrida Beach (49.2%) compared to 22.4% on Ploče Beach. At the same time, the number of bathers on the beaches indicated greater pressure on Kantrida Beach. It could be assumed that, in addition to higher sampling frequency, the pressure exerted by more bathers also accounted for the higher share of *S. aureus* positive samples, suggesting that bathers are a possible source of *S. aureus* contamination. This is supported by the share of *S. aureus* positive samples on Kantrida and Ploče beaches (36.3% or 90 out of 248 samples), compared to the data obtained by the official national monitoring program at all bathing sites in the county (2.2% or 57 out of 2619 samples). A plausible reason is the difference in beach load, i.e., number of bathers. Kantrida and Ploče are two of the most frequented beaches of the county and considerably more loaded with bathers than other beaches. A significant correlation between the presence of *S. aureus* in seawater and the number of bathers in the coastal environment has been shown by Cheung et al. [31] and Yoshpe-Purer and Golderman [10], who reported a significantly higher incidence of *S. aureus* in Mediterranean coastal waters during the peak bathing period at the most frequented beaches (91%) compared to the less frequented ones (49.5%).

Bathers were confirmed as the possible main source of *S. aureus* by a relatively weak but statistically significant positive correlation between *S. aureus* counts and the number of bathers (rs = 0.321; *p* < 0.05). Higher levels of *S. aureus* are probably due to the higher number of bathers. Similar findings were recorded by Šolić and Krstulović [32], who noted that the highest concentrations were found on the extremely crowded beaches of Split (Croatia). Other studies also found a significant correlation between *S. aureus* levels in seawater and the number of bathers determined by observation at the time of sampling [20,33,34]. 

An analysis of the results of supplemental monitoring revealed a positive correlation between *S. aureus* counts, seawater and air temperature, number of bathers, and number of children in diapers (Table 1). Since the variations in water temperature were relatively low (2 °C) (Table 2), temperature could not affect the differences in bacterial counts considerably. It is very likely that the only causal relationship is that between the number of bathers and *S. aureus* in seawater. Other correlations are likely artefacts, without causal relationship. In other words, during sunny periods, both air temperature and seawater temperature increase, causing favorable meteorological conditions that attract more beach goers to the beaches, and consequently more children in diapers. That means that temperature did not affect the levels of *S. aureus* directly but indirectly by attracting a larger numbers of bathers to the beaches. Goodwin et al. [20] also reported a positive correlation between *S. aureus* and seawater temperature but since they did not examine the relationship between temperature and the number of bathers, the data is insufficient to conclude on whether this is an actual correlation or not.

A primary study of *S. aureus* in avian droppings proved that bird droppings can be an important reservoir of this pathogen [21]. Cragg and Clayton [35] reported that *S. aureus* is one of the commonest bacteria in the fecal flora of seagulls. In addition to birds, domestic animals, including dogs, can also carry this opportunistic pathogen [19,36]. Hence, beaches with more seagulls and dogs may be considered as those with a higher risk of *S. aureus* contamination. Since no correlation between *S. aureus* and number of seagulls and dogs at the beach was found in this study, seagulls and dogs can be excluded as likely sources of this bacteria on the studied beaches.

Although some studies recorded a significant correlation between *S. aureus* and FIB for moderate and extremely polluted areas [32,37], this study did not show a significant correlation between these two parameters. Other studies also reported the absence of any correlation between FIB and *S. aureus* on highly frequented beaches. A staphylococcal impetigo outbreak in the very popular and frequently visited tourism resort of Vodice in Croatia (July of 2015, in the midst of the bathing season) was not accompanied by increased FIB levels. In 25% of the samples taken from beaches in the Vodice area during and after the outbreak, *S. aureus* counts in seawater were >150 CFU/100 mL, while bathing quality was excellent, with very low or undetectable FIB levels [38]. In a study carried out in Egyptian coastal waters, 35% of bathing water samples exceeded the established guideline values for *S. aureus*, without any of indicator bacteria being detected [33].

Unlike *S. aureus*, both indicator bacteria, *E. coli* and intestinal enterococci, correlated negatively with the number of bathers. This could indicate that FIB at these sites may not have originated primarily from bathers but from other plausible sources such as coastal springs, seagulls, and dogs, while *S. aureus* originates from non-fecal sources predominantly shed from the skin and possibly anterior nares of bathers, as reported by Elmir et al. [22].

Furthermore, in addition to attracting more people to the beach, warm and sunny periods undoubtedly resulted in a higher reduction of indicator bacteria counts in seawater. FIB concentration is greatly influenced by the weather and environmental conditions [32,39,40]. It is well-known that an unfavorable marine environment, particularly solar radiation, temperature, and salinity, have a negative effect on allochthones bacteria survival, reducing culturability of FIB in a very short period of exposure [41,42,43,44], considerably more intense than the culturability of *S. aureus* [32]. A greater resistance of *S. aureus* to marine conditions, and consequently longer survival compared to FIB, is a result of better tolerance for higher salinity [25]. Lower enterococci levels compared to *S. aureus* were recorded in a study by Enns, et al. [7]. This was attributed to a combination of the different levels shed by humans coupled with possible higher sensitivity to solar radiation. Better resistance to environmental factors might be attributed to the clustered structure of *S. aureus*, which reduces the area exposed to environmental factors, solar radiation in particular.

The awareness of bathers also contributes to the negative correlation between FIB counts and the number of bathers. Beachgoers are most likely even more aware of the importance of the quality of bathing water and they regularly check the quality of bathing water before going to the beach. They avoid visiting a beach when the water is of poorer quality. This assumption is supported by a significant increase of web access statistics (unique Hit Counters by location) for web application “Sea bathing water quality on beaches”, from 13,881 in 2009 to 56,119 for Primorje-Gorski Kotar County in the 2019 bathing season [45].

Unlike *S. aureus*, both indicator bacteria were significantly positively correlated with the number of seagulls and dogs. Besides dog faces and seagull droppings being possible direct sources of fecal contamination, weather conditions likely contributed to higher FIB counts. Generally, cloudy weather and rainy periods contribute to lower FIB reduction in seawater, by lower solar radiation, lower air and seawater temperature and salinity. All these parameters correlated negatively with FIB counts. Since no correlation between the number of bathers and FIB bacteria was found, numerous coastal and underwater springs that intensify after rain events, are the most likely source of fecal contamination at these beaches. This is confirmed by lower salinity and sea temperature, and higher turbidity recorded at Kantrida beach, with pronounced variations in salinity values, compared to Ploče beach. Thus, the levels of *E. coli* and intestinal enterococci were higher at Kantrida beach. Mance et al. [46] also reported coastal springs on Kantrida Beach as occasional sources of higher microbial load.

Finally, based on the assessment of bathing water quality at bathing sites included in supplemental monitoring, it could be concluded that FIB are not always a good indicator of *S. aureus* incidence and levels. Consequently, bathing waters can pass the FIB standard but still have high counts of *S. aureus*. This is supported by the fact that all bathing sites, where supplemental monitoring was carried out, were assessed as having excellent or good quality (Figure 4). It means that there was a low chance of the occurrence of pathogenic microorganisms and low risk of waterborne diseases associated to bathing. Furthermore, in as many as 20% of samples positive for *S. aureus*, the levels of at least one FIB were lower than the levels of *S. aureus*. Additionally, in a few *S. aureus* positive samples, no FIB was detected. Unfortunately, there is no data on possible bathing-related infections reported by bathers on these beaches, so it cannot be argued whether the recorded levels of *S. aureus* resulted in an increased incidence of staphylococcal infections. In general, there is a lack of studies addressing the relationship between recorded *S. aureus* counts in bathing water and the occurrence of symptoms of staphylococcal infection. One of the reasons for this could be a general failure of beach-goers to report some of the symptoms due to their mild nature and short duration [47]. The main finding by the retrospective epidemiological monitoring study conducted by Charoenca and Fujioka [48] was the strong association between marine water contact and staphylococcal infections. Water analyzes were performed subsequently, so it was not possible to determine *S. aureus* levels in the water at the time of exposure, but *S. aureus* isolates recovered from skin and water had similar antibiotic sensitivity patterns and phage typing, supporting the conclusion that marine waters were the transmission medium for staphylococcal infections.

Furthermore, available scientific literature does not provide evidence for a limit value based on dose-response relation for exposure to *S. aureus* in bathing water [49]. In some studies, authors used the value of 100 CFU/100 mL, proposed by Favero et al. [50] as the permissible maximum for swimming pool water. Considering that the study included staphylococci in general, not just *S. aureus*, the proposed value cannot simply be applied to *S. aureus*. Therefore, further studies are needed to examine which levels of *S. aureus* in bathing water may lead to the onset of symptoms of staphylococcal infections and to determine appropriate limit values. A faster procedure and more selective method, without additional confirmatory tests, would also contribute to better and more reliable data as well as to more timely results.

However, it seems that there is a justified need for the inclusion of additional parameters, such as *S. aureus*, in the monitoring of bathing water. In addition, regional specificities should also be taken into account when revising the Directive. The EU is a vast area with diverse geographical, climatic, and hydrological characteristics that may influence the occurrence and persistence of certain allochthones microorganisms, such as *S. aureus,* and naturally occurring microorganisms, such as *P. aeruginosa* and *Vibrio* species, as well as natural phenomena such as cyanobacterial blooms. This may affect the suitability of the current indicators as the only indicators of the microbiological quality of bathing water. For this reason, the Directive should allow Member States to include additional, regionally specific parameters in their national bathing water quality management legislation. This would significantly improve the protection of human health, which is the main purpose of the Directive.

## 5. Conclusions

The correlation between the number of bathers on a beach and *S. aureus* counts in the seawater samples indicates that the contamination of Kantrida and Ploče beaches by *S. aureus* is likely due to bathers. When compared to other, less-loaded beaches in the county, the incidence and levels of *S. aureus* were significantly higher at these urban beaches, very often overloaded with bathers. This was probably not only due to the higher number of bathers, as a main source of *S. aureus*, but also to significantly higher sampling frequency, which increases the probability of recording pollution. No correlation was found between *S. aureus* and fecal indicator bacteria (FIB), *E. coli* and intestinal enterococci, thus indicating that *S. aureus* was not of fecal origin and that the current indicator bacteria are not always an adequate indicator of health risk.

Based on its incidence and levels recorded in seawater in this study, the possibility of including *S. aureus* as an additional parameter in official (routine) monitoring of bathing water quality at excessively loaded urban beaches is justified, regardless of the microbiological seawater quality based on FIB levels. Sampling frequency at these beaches should also be increased. Additional research is required in order to obtain a better understanding of the issue and document relevant discussions. In addition to conducting epidemiological studies regarding the correlation of *S. aureus* in bathing water with the incidence of staphylococcal infections, current sampling frequency should be reconsidered, and guideline values of *S. aureus* need to be proposed.

## Figures and Tables

**Figure 1 ijerph-18-05234-f001:**
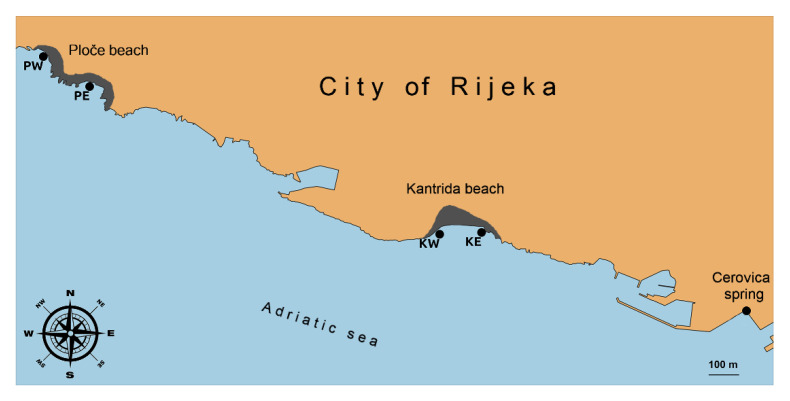
Kantrida (east-KE, west-KW) and Ploče (east-PE, west-PW) beaches and sampling sites.

**Figure 2 ijerph-18-05234-f002:**
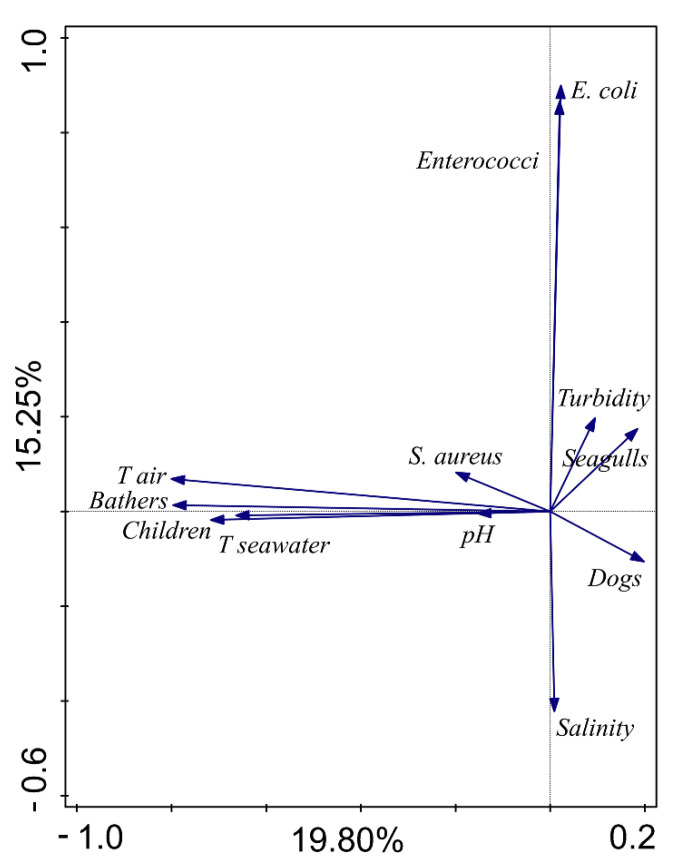
PCA analysis biplot.

**Figure 3 ijerph-18-05234-f003:**
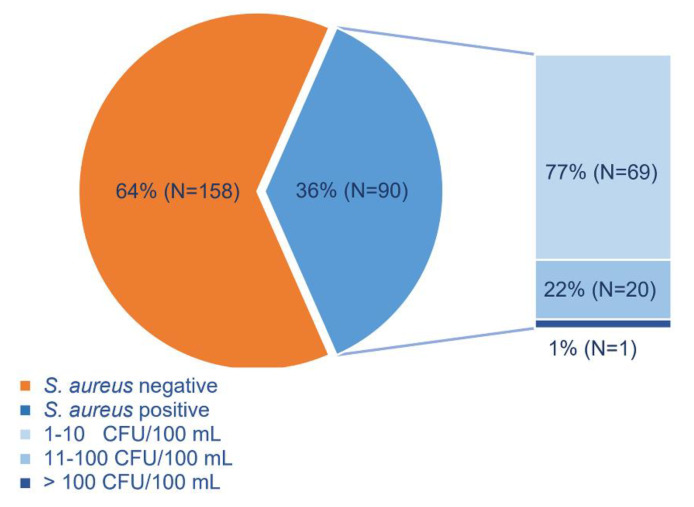
Percentage of *S. aureus* positive seawater samples, based on *S. aureus* level.

**Figure 4 ijerph-18-05234-f004:**
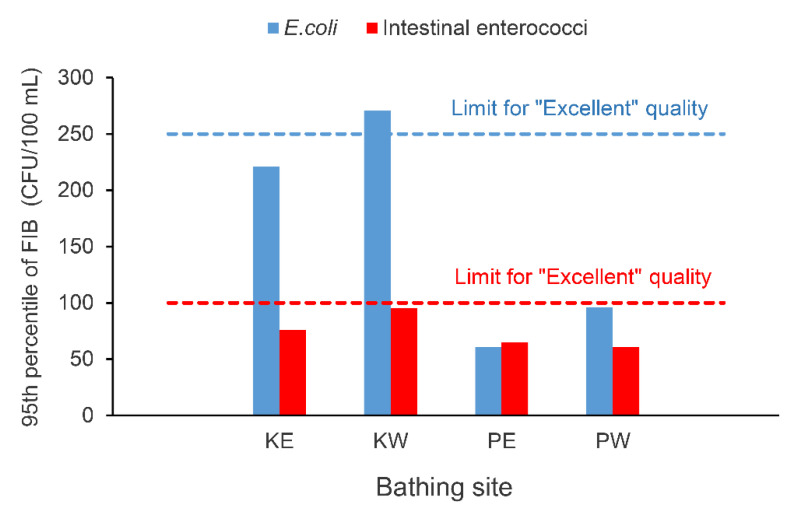
The 95th percentile of FIB data obtained by supplemental monitoring.

**Table 1 ijerph-18-05234-t001:** Spearman correlation coefficients between the examined variables (supplemental monitoring data). Statistically significant correlations (*p* < 0.05) are highlighted in bold.

Variables	*S. Aureus* (CFU/100 mL)	Salinity	ph	Turbidity (NTU)	*E. coli* (CFU/100 mL)	Enterococci (CFU/100 mL)	Air Temperature (°C)	Water Temperature (°C)	*N* Bathers	*N* Children in Diapers	*N* Dogs
Salinity	−0.033										
pH	0.048	−0.020									
Turbidity (NTU)	0.046	−**0.353**	−0.025								
*E. coli* (CFU/100 mL)	0.084	−**0.395**	−0.094	0.211							
Enterococci (CFU/100 mL)	0.088	−**0.379**	−0.057	**0.208**	**0.695**						
Air temperature (°C)	**0.147**	**0.111**	0.052	0.008	−**0.273**	−**0.276**					
Water temperature (°C)	**0.243**	0.066	**0.274**	−**0.252**	−**0.319**	−**0.307**	**0.416**				
*N* bathers	**0.321**	**0.131**	0.081	−**0.226**	−**0.136**	−**0.161**	**0.367**	**0.498**			
*N* children in diapers	**0.203**	**0.117**	0.014	−**0.130**	−**0.220**	−**0.176**	**0.355**	**0.395**	**0.673**		
*N* dogs	−0.065	−**0.286**	−0.066	**0.287**	**0.294**	**0.264**	−**0.341**	−**0.419**	−**0.390**	−**0.261**	
*N* seagulls	0.035	−**0.208**	0.039	**0.221**	**0.316**	**0.216**	−**0.166**	−**0.221**	−**0.229**	−**0.382**	**0.252**

**Table 2 ijerph-18-05234-t002:** Values of parameters measured at Kantrida and Ploče beaches.

Parameter	Median (IQR 25–75)	
	Ploče	Kantrida
*S. aureus* (CFU/100 mL)	2 (1–3)	6 (2–14)
*E. coli* (CFU/100 mL)	7 (3–22)	29.5 (11.5–77)
Intestinal enterococci (CFU/100 mL)	7 (3–18)	18 (10–35)
Salinity	36.1 (34.2–37.2)	35.0 (33.0–36.7)
Water temperature (°C)	25.0 (23.0–25.0)	24.0 (23.0–25.0)
Air temperature (°C)	26.0 (25.0–28.0)	26.0 (25.0–28.0)
Turbidity (NTU)	0.50 (0.38–0.7)	0.62 (0.49–0.85)
pH	8.0 (7.9–8.0)	8.0 (7.9–8.0)

## Data Availability

The data presented in this study are available on request from the corresponding author. The data are not publicly available due to the fact that large portion derives from the official Croatian national monitoring. In that regard, raw data about the quality of the bathing seawater are not published on the official platform, and only the categories of bathing water quality (excellent, good, satisfactory and poor) are issued. The reason for that is inadequate level of education of the public for interpretation of gathered raw bacterial counts in seawater.

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
