# Peer review of "Staphylococcus aureus—An Additional Parameter of Bathing Water Quality for Crowded Urban Beaches"

_ijerph, 2021, doi:10.3390/ijerph18105234_

Round 1

Reviewer 1 Report

The article describes a study that seeks to incorporate the detection of S. aureus into the official monitoring parameters, to establish the quality of bathing water in urban beaches. Although the detection of S. aureus is important due to the bibliography presented, the main weakness of the study is the lack of causality between the recorded levels of S. aureus and staphylococcal infections. This reviewer suggests that the authors discuss this relationship, describing more studies in geographic areas with similar characteristics.

In addition, it is recommended that you improve Figure 1, presenting it with more information, such as sampling locations within the indicated areas, average temperature, water temperature and number of bathers per area.

Reviewer 2 Report

The presented manuscript is of excellent quality. I found it highly interesting and easy to read and follow. I appreciate intense sampling effort made to demonstrate the possible non-detected outbreaks of the pathogen. The estimated concentrations will be helpful to determine the beground levels of S. aureus in the environment.

To strengthen the analytical part and main conclusion about ‘internal’ and ‘external’ sources of pathogene contamination, I would recommend to apply multivariate analyses methods next to multiple rank correlations matrix. There is relatively high correlation between two intestinal pathogens, while S. aureaus has rather different pattern of variation, the set of favorable factors is different, or even remains still unknown.

Another suggestion for the discussion part is to add a final conclusion referring to BW regulations at the EU level and country level. What this study adds to the ongoing debate about too few parameters considered in EU BWD. Some scientists suggest to add Vibrion counts, some cyanobacteria counts. What could be further arguments selecting S. aureus as candidate indicator among other emerging pathogens?

Reviewer 3 Report

Although the Bathing Water Directive does not include other microbial parameters, the new Drinking Water Directive and the Reclaimed Water Directive include the analysis of somatic coliphages which persist longer in the environment. For this reason, I miss some information of the S. aureus persistence in marine waters compared to other microbial parameters that persist longer in the environment such as the somatic coliphages and spores of sulphite-reducing clostridia.

The manuscript seems almost suitable for publication, just needs some corrections, please see the comments below:  

Lines 17-18: rewrite this sentence. Change for something like: During the last years, the report of the occurrence of waterborne disease symptoms related to non-enteric pathogens has increased, without any record of higher levels of indicator bacteria (Escherichia coli and intestinal enterococci).

Line 22: remove “(S. aureus)”.

Line 38: remove “(E. coli)”.

Line 57: remove “(S. aureus)”.

Line 101: Early in the morning? Depending on time, you could be analysing the microorganisms released in the previous day and the difference in their persistence could determine the results.

Line 203: add intestinal before enterococci.

Line 255: Clarify if the routine sampling was biweekly (twice per week) or fortnightly (once every two weeks) as you say in the M&M section.

Lines 263-265: Please, rewrite this sentence.

Line 322: add intestinal before enterococci.

Table 2: Add the salinity unites.

Figure 2: Indicate the number of the positive samples out of the total number of samples.
